# Canadian Expert Consensus Recommendations for the Diagnosis and Management of Glioblastoma: Results of a Delphi Study

**DOI:** 10.3390/curroncol32040207

**Published:** 2025-04-01

**Authors:** Warren P. Mason, Rebecca A. Harrison, Sarah Lapointe, Mary Jane Lim-Fat, Mary V. MacNeil, David Mathieu, James R. Perry, Marshall W. Pitz, David Roberge, Derek S. Tsang, Christina Tsien, Frank K. H. van Landeghem, Gelareh Zadeh, Jacob Easaw

**Affiliations:** 1Department of Medicine, Temerty Faculty of Medicine, University of Toronto, Toronto, ON M5G 2M9, Canada; 2Department of Medicine, University of British Columbia, Vancouver, BC V5Z 4E6, Canada; 3Department of Medicine, Centre Hospitalier Universitaire de Montreal, Montreal, QC H2X 3J4, Canada; 4Faculty of Neuroscience, University of Montreal, Montreal, QC H3T 1J4, Canada; 5Department of Medicine, Sunnybrook Health Sciences Centre, University of Toronto, Toronto, ON M4N 3M5, Canada; 6Department of Medicine, Dalhousie University, QE II Health Science Centre, Halifax, NS B3H 2Y9, Canada; 7Department of Medicine, Nova Scotia Cancer Care, Halifax, NS B3H 1V8, Canada; 8Department of Surgery, Université de Sherbrooke, Sherbrooke, QC J1H 5N4, Canada; 9Odette Cancer Centre, Sunnybrook Health Sciences Centre, Toronto, ON M4N 3M5, Canada; 10Department of Medicine, University of Toronto, Toronto, ON M5S 1A4, Canada; 11Department of Internal Medicine, University of Manitoba, Winnipeg, MB R3E 0V9, Canada; 12Medical Oncology and Hematology, CancerCare Manitoba, Winnipeg, MB R3E 0V9, Canada; 13Division of Radiation Oncology, Centre Hospitalier Universitaire de Montreal, Montreal, QC H2X 0C1, Canada; 14Department of Radiology, Radiation-Oncology and Nuclear Medicine, University of Montreal, Montreal, QC H3T 1J4, Canada; 15Department of Radiation Oncology, Princess Margaret Cancer Centre, Toronto, ON M5G 2M9, Canada; 16Department of Radiation Oncology, McGill University, Montreal, QC H4A 3J1, Canada; 17Department of Laboratory Medicine and Pathology, University of Alberta, Edmonton, AB T6G 2B7, Canada; 18Department of Surgery, University of Toronto, Toronto, ON M5T 1P5, Canada; 19Krembil Brain Institute, University Health Network, Toronto, ON M5T 1M8, Canada; 20Department of Oncology, University of Alberta, Edmonton, AB T6G 1Z2, Canada

**Keywords:** glioblastoma, diagnosis, management, molecular markers, targeted therapy, immunotherapy, tumor-treating fields

## Abstract

Glioblastoma is the most common and aggressive malignant brain tumor in adults, with an increasing incidence and a poor prognosis. Current challenges in glioblastoma management include rapid tumor growth, limited treatment effectiveness, high recurrence rates, and a significant impact on patients’ quality of life. Given the complexity of glioblastoma care and recent advancements in diagnostic and treatment modalities, updated guidelines are needed in Canada. This Delphi study aimed to develop Canadian consensus recommendations for the diagnosis, classification, and management of newly diagnosed and recurrent glioblastoma. A multidisciplinary panel of 14 Canadian experts in glioblastoma care was convened, and a comprehensive literature review was conducted to synthesize evidence and formulate initial recommendations. Consensus was achieved through three Delphi rounds, in which panelists rated their agreement with recommendation statements on a five-point Likert scale. Statements with ≥75% agreement were accepted, and others were revised for re-voting. Final recommendations were formulated based on the consensus level, strength of evidence, clinical expertise, and consideration of the Canadian healthcare context. These recommendations aim to standardize glioblastoma diagnosis and classification across Canada, provide evidence-based guidance for optimal treatment selection, integrate novel therapies, and enhance the overall quality of care for glioblastoma patients.

## 1. Introduction

Glioblastoma is the most prevalent malignant tumor of the central nervous system (CNS) [1], representing approximately half of all CNS malignant tumors [2]. In Canada between 2013 and 2017, the average annual age-standardized incidence (ASIR) of glioblastoma ranged from 0.22 per 100,000 (95% CI, 0.17–0.28) in individuals aged between 0 and 19 to 13.40 per 100,000 (95% CI, 12.91–13.90) in adults aged 65 years or older [2].

Glioblastoma is an aggressive and heterogeneous malignancy that causes significant morbidity and mortality. The median survival time of patients diagnosed with glioblastoma between 2010 and 2017 in Canada was only 8.1 months, and the 5-year overall survival rate was 4.9%, which is the lowest among all patients with CNS tumors [1]. Current challenges in glioblastoma diagnosis and management that may contribute to the poor prognosis of the disease include the rapid growth and high invasiveness of glioblastomas [3], the limited effectiveness of current treatments [3], the development of treatment resistance [4], near-invariable recurrence [5], difficulty in assessing the treatment response because of pseudo-progression [6,7,8], and a significant impact on patient’s quality of life and cognitive functions [9,10].

Given the complexity of glioblastoma management and the rapid evolution of diagnostic and treatment modalities, there is a need for up-to-date, evidence-based guidelines tailored to the Canadian healthcare landscape. The latest Canadian guidelines for the management of glioblastoma were published in 2011 [11,12], and significant advancements have since been made. These advancements include the integration of molecular markers into diagnostic and prognostic algorithms [13,14,15], improvements in surgical techniques and imaging technologies [16,17,18], and the introduction of novel therapies such as tumor-treating fields (TTFields) [19].

The objective of this Delphi study was to develop updated Canadian recommendations for the diagnosis, classification, and management of newly diagnosed and recurrent glioblastoma. These recommendations aim to standardize glioblastoma diagnosis and classification across Canada, provide evidence-based guidance for optimal treatment selection, address the integration of novel therapies and technologies into clinical practice, and enhance the overall quality of care for patients with glioblastoma in Canada.

## 2. Materials and Methods

### 2.1. Expert Panel Selection and Composition

A multidisciplinary panel of 14 Canadian experts in glioblastoma management was convened. The panel included five neuro-oncologists, two neurosurgeons, three radiation oncologists, one neuropathologist, and three medical oncologists (Appendix A). The experts were selected based on their clinical experience, research contributions, and geographic representation across Canada.

### 2.2. Literature Review

The PubMed, Embase, Google Scholar, and Cochrane Library databases were searched for relevant articles published between January 2010 and May 2024. Search terms included glioblastoma, high-grade glioma, diagnosis, treatment, molecular markers, surgery, radiotherapy, chemotherapy, targeted therapy, immunotherapy, and TTFields. The selected studies included randomized controlled trials, systematic reviews, and cohort or case series. High-quality evidence (meta-analyses and phase 3 randomized trials) was favored whenever possible. Additional sources of evidence for drafting the initial statements included Canadian and international guidelines, conference abstracts, and ongoing clinical trials. A list of all the research articles, guidelines, and conference abstracts is provided in Appendix A. Data extracted from the studies were used to generate tables describing the study design, quality, and outcomes (Appendix A). These tables were used to generate evidence-based recommendations.

### 2.3. Delphi Study Design and Consensus Process

After the literature and existing guidelines were reviewed, the co-chairs synthesized the evidence and formulated 50 initial recommendations for the diagnosis and management of glioblastoma. Both the level of evidence and the quality of the studies were considered for the formulation of the initial recommendations. Consensus was achieved through three Delphi rounds (Figure 1). During Round 1, panel members were presented with initial recommendation statements based on the literature review and were asked to rate their agreement on a 5-point inverted Likert scale (1 = strongly agree, 5 = strongly disagree). In addition to rating their agreement with the statements, the panelists used a GRADE rating system (Appendix A) to rate the quality of supporting evidence. They were also encouraged to provide comments and suggestions for modification. During Round 2, revised statements incorporating feedback from Round 1 were circulated, including feedback from a virtual meeting in which all panelists were invited. The panel members re-rated their agreement and provided final comments. A final Round 3 consensus meeting was held to review, revise, and reevaluate recommendations that did not reach consensus during Round 2.

Consensus was defined as ≥75% of participants rating a statement as 1–2 on the inverted Likert scale (strongly agree or agree). Statements that did not meet these criteria were revised and re-voted upon. Any areas of disagreement were resolved through discussions among all panel members.

Final recommendations were formulated based on the consensus level from the Delphi process, the strength of supporting evidence, the clinical expertise and judgment of the panel, and a consideration of the Canadian healthcare context.

## 3. Results

### 3.1. Literature Search Results, Initial Recommendations, and Consensus Process

The search retrieved 123 papers (Appendix A), including three guidelines, that were used for discussion and the formation of the initial evidence-based recommendations by the co-chairs. In total, 12 panel members participated in Round 1 of the Delphi process (86% participation rate), 13 members participated in Round 2 (93% participation rate), and 10 members participated in Round 3 (71% participation rate). At the end of the process, 45 (87%) statements reached a consensus, and 7 (13%) did not.

### 3.2. Recommendations

#### 3.2.1. Glioblastoma Diagnosis and Classification

The panel reached a consensus on eleven statements related to glioblastoma diagnosis and classification (Table 1; Figure 2A). The recommendations centered on imaging (*n* = 3), tumor classification (*n* = 3), and molecular profiling (*n* = 5). Eight (72%) of these recommendations reached a consensus during Round 1 of the Delphi process.

#### 3.2.2. Management of Newly Diagnosed Glioblastoma

The panel reached a consensus on 14 statements related to the management of newly diagnosed glioblastoma (Table 2; Figure 2B). Eight (57%) of these recommendations reached a consensus during Round 1 of the Delphi process, four (29%) reached a consensus during Round 2, and two (14%) during Round 3.

#### 3.2.3. Monitoring Treatment Response and Disease Progression

The panel reached a consensus on six statements related to the monitoring of treatment response and disease progression (Table 3; Figure 2C). Five (83%) of these recommendations reached a consensus during Round 1 of the Delphi process and one (17%) during Round 3.

#### 3.2.4. Management of Recurrent or Progressive Glioblastoma

The panel reached a consensus on nine statements related to the management of recurrent or progressive glioblastoma (Table 4; Figure 2D). The recommendations centered on radiotherapy, chemotherapy, and TTFields (*n* = 3), surgery (*n* = 3), personalized treatment approaches (*n* = 1), clinical trials (*n* = 1), and targeted therapy and immunotherapy (*n* = 1). Eight (89%) of these recommendations reached a consensus during Round 1 of the Delphi process, and one (11%) reached a consensus during Round 2.

#### 3.2.5. Supportive Care

The panel reached a consensus on five statements related to supportive care for glioblastoma (Table 5; Figure 2E). Four (80%) of these recommendations reached a consensus during Round 1 of the Delphi process, and one (20%) reached a consensus during Round 2.

#### 3.2.6. Statements That Did Not Reach Consensus

Seven statements (13%) did not reach consensus (Table 6). The consensus levels for these statements ranged from 23% to 67%. The largest disagreement was related to prophylaxis for pneumocystis pneumonia in patients receiving chemoradiotherapy or adjuvant chemotherapy with temozolomide (consensus level: 23%) and venous thromboembolism (VTE) prophylaxis with low-molecular-weight heparin after surgical tumor resection (consensus level: 46%).

## 4. Discussion

### 4.1. Glioblastoma Diagnosis and Classification

#### 4.1.1. Imaging

The timely diagnosis of glioblastoma is crucial due to the rapid growth and infiltration of surrounding brain tissues. Diagnostic workup typically includes medical history, physical examination, and neurological examination (including cranial nerve, motor, sensory, reflex, coordination, and gait testing, in addition to attention, judgment, memory, and speech) [20]. The panelists reached a strong consensus that all patients with a suspected diagnosis of glioblastoma should undergo magnetic resonance imaging (MRI) of the brain and that computed tomography (CT) should be used to evaluate patients with suspected glioblastoma only if MRI is unavailable or contraindicated. Although CT is often the first imaging modality used to detect or rule out the presence of brain tumors, MRI is more sensitive and the preferred test for further evaluation of suspicious brain lesions [21]. It was highlighted that CT may be useful as part of the evaluation of a patient with a known glioblastoma in cases of clinical worsening, hemorrhage, or increased edema.

A moderate consensus was achieved regarding the use of advanced brain MRI techniques, such as diffusion-weighted MRI and perfusion-weighted MRI, positron emission tomography (PET), and magnetic resonance (MR) spectroscopy, to assess the presence of glioblastoma in difficult cases or to help distinguish glioblastoma from other tumor types and progression from radionecrosis. The results of a recent meta-analysis of 27 imaging studies support the use of brain MRI with T2, fluid-attenuated inversion recovery (FLAIR), and pre- and post-contrast T1 sequences as the minimum assessment for suspected glioblastoma [22]. Evidence also suggests that the addition of diffusion- and perfusion-weighted MRI, MR spectroscopy, and PET imaging could enhance the diagnostic specificity and prognostic value of imaging for glioblastoma and could be used to assess the presence of glioblastoma in difficult cases (such as in individuals with intracranial hemorrhage) [21,22,23].

#### 4.1.2. Tumor Classification

Glioblastoma classification has evolved with the incorporation of molecular and genetic features into traditional histological criteria, and molecular testing is increasingly being used in glioblastoma diagnosis, prognostication, treatment response prediction, and clinical decision-making. Mutations in the gene encoding isocitrate dehydrogenase (IDH) have been used for the molecular classification of glioblastoma and other types of gliomas. For many years, glioblastoma has been classified as IDH-mutant glioblastoma or IDH-wildtype glioblastoma [24]. Under that classification, approximately 90–95% of grade 4 astrocytoma have no *IDH1/2* mutations, while the remaining 5–10% harbor *IDH1/2* mutations [4]; however, the latter are now classified as IDH-mutant astrocytoma, WHO grade 4, according to the 2021 World Health Organization (WHO) classification criteria [13,14]. In accordance with the 2021 WHO classification criteria, the panelists reached a strong consensus that only IDH-wildtype grade 4 gliomas should be classified as glioblastoma and that IDH-wildtype astrocytic gliomas in adults should be considered glioblastoma if any of the following criteria are met: microvascular proliferation, necrosis, *TERT* promoter mutation, *EGFR* gene amplification, and +7/−10 chromosome copy number changes. However, one expert noted that *TERT* promoter mutations are not necessarily prognostic [25], and another panelist highlighted that they are not tested in all cases.

#### 4.1.3. Molecular Profiling

Panelists reached a strong consensus that all patients with newly diagnosed gliomas should be tested (by immunohistochemistry or sequencing) for *IDH* mutations. This recommendation aligns with the 2021 WHO criteria [13,14] and the Canadian Association of Neuropathologists recommendations for the molecular testing of CNS tumors [26]. In addition to facilitating accurate classification, testing for *IDH* mutations also provides prognostic information, as *IDH1/2* mutations have been associated with an improved prognosis in patients with high-grade gliomas [15]. According to recent retrospective data, the median overall survival (OS) of patients with IDH-mutant astrocytoma is 78.5–161.0 months, whereas the median OS of patients with IDH-wildtype glioblastoma is only 22.0–27.2 months [27].

The panelists agreed that, in patients with confirmed glioblastoma (IDH-wildtype) or grade 4 IDH-mutated astrocytoma, *MGMT* promoter methylation status should be tested to inform prognosis and predict response to temozolomide chemotherapy. Approximately 40–60% of glioblastomas show high methylation levels in the promoter of *MGMT*, the gene encoding O6-methylguanine-DNA methyltransferase [4,28]. *MGMT* promoter methylation inactivates the expression of *MGMT* and has been associated with longer patient survival and tumor response to temozolomide chemotherapy in patients with glioblastoma [29,30,31]. According to data from a randomized phase 3 trial involving 573 patients with newly diagnosed glioblastoma who received radiotherapy alone or with concomitant and adjuvant temozolomide, *MGMT* promoter methylation is a strong predictor of benefit from temozolomide chemotherapy [32].

Finally, the panelists agreed that molecular profiling may reveal genetic abnormalities that may alter the diagnosis or provide actionable treatment options, which aligns with the growing body of evidence supporting the relevance of marker identification in glioblastoma management [33]. These additional markers include *EGFR* amplification and chromosome 7 gains and 10 losses. *EGFR* gene amplification is found in approximately 40–60% of all glioblastomas, leading to higher EGFR expression and poor prognosis [4]. The loss of heterozygosity (LOH) on chromosome arm 10q is also common in glioblastoma, occurring in more than 50% of cases [34]. *ADD3* has been identified as a novel tumor suppressor located on chromosome arm 10q, and it may link LOH on chromosome 10q to glioblastoma development and progression [34]. The fact that these alterations cannot discriminate glioblastomas from other brain tumors and the lack of high-quality evidence to support their use in clinical decision-making have contributed to uncertainty among panelists regarding the clinical usefulness of testing for these molecular alterations.

### 4.2. Management of Newly Diagnosed Glioblastoma

#### 4.2.1. General

There are no curative treatments for glioblastoma, and its management is geared toward slowing disease progression, alleviating symptoms, and improving quality of life. Considering that the intent of management is palliative or non-curative from the outset, the panelists agreed that part of the discussion on further management should involve conversations with the patient and their family about the goals of care and whether they wish to proceed with active therapies. For some patients with a very poor performance status, palliative care alone is a reasonable option.

In accordance with previous Canadian guidelines for the management of glioblastoma [12], the panelists recommend that, when deciding the first-line treatment for patients with glioblastoma, clinicians must take into account the tumor characteristics (size, location, and integrated diagnosis) and the patient’s age, functional ability measured as Karnofsky performance status (KPS), symptoms, clinical needs, and preferences. Multidisciplinary care teams would optimally include neurosurgeons, medical oncologists or neuro-oncologists, radiation oncologists, neuropathologists, radiologists, and neurologists.

#### 4.2.2. Surgery

In accordance with the most recent Canadian guidelines for the management of glioblastoma [12], the panelists agreed that all patients with newly diagnosed operable glioblastoma and those with good performance status should undergo maximal safe resection. In cases not amenable to resection (i.e., poor performance status and those with unfavorable tumor location), stereotactic biopsy is recommended to establish a histopathologic diagnosis and molecular profile (including cytogenetic alterations). The panelists also agreed that tissue should be saved for tumor banking whenever feasible. Two large meta-analyses involving more than 41,110 patients with glioblastoma showed that, compared with subtotal resection (STR), gross total resection (GTR) was significantly associated with improved 1-year and 2-year OS and progression-free survival (PFS) [16,17]. Compared with STR, GTR provided a relative risk (RR) for OS of 0.62 (95% confidence interval [CI], 0.56–0.69; *p* < 0.001) at 1 year and 0.84 (95% CI, 0.79–0.89; *p* < 0.001) at 2 years. Similarly, the RR of disease progression with GTR compared to STR was 0.72 (95% CI, 0.48–1.09; *p* = 0.12) at 6 months and 0.66 (95% CI, 0.43–0.99; *p* < 0.001) at 1 year. A retrospective analysis of data from 64 patients with newly diagnosed glioblastoma demonstrated that the median OS was 24 months with supratotal resection, 16 months with GTR, 14 months with STR, and 10 months with biopsy (Appendix A) [35]. The median PFS was 12.6, 10, 5.9, and 3.2 months, respectively [35]. Although there is no evidence from randomized studies, the consensus amongst most neuro-oncologists and surgeons is that maximal safe surgery is the accepted standard of care.

#### 4.2.3. Radiotherapy and Chemotherapy

The panelists reached a consensus that radiation therapy should be initiated as soon as it is safely permissible, ideally within 3–6 weeks after surgery, using gadolinium-enhanced T1-weighted MRI and FLAIR to determine the clinical target volume (CTV). In a study of 26 patients with newly diagnosed glioblastoma, radiotherapy initiated within 6 weeks after surgery provided better OS, and PFS than radiotherapy initiated more than 6 weeks after surgery (median OS: 26.6 [95% CI, 18.3–34.9] vs. 15.7 [95% CI, 9.2–22.3] months, *p* = 0.001; median PFS: 16.3 [95% CI, 14.7–18.0] vs. 9.1 [95% CI, 4.7–13.4] months, *p* = 0.006) [36].

Consensus was reached regarding the use of 75 mg/m^2^ concurrently with 60 Gy radiotherapy for 42 days, followed by adjuvant temozolomide 150–200 mg/m^2^ in a 5/28-day schedule for six cycles in patients aged ≤ 70 years with good performance status (KPS ≥60). Data from the phase 3 EORTC-NCIC randomized trial suggest that adding chemotherapy (adjuvant temozolomide) to radiotherapy after surgery may increase survival and quality of life for patients with newly diagnosed glioblastoma [32,37]. At two years, the overall survival was 14.6 months (95% CI, 13.2–16.8) with radiotherapy plus temozolomide compared to 12.1 months (95% CI, 11.2–13.0) with radiotherapy alone [37]. Additionally, temozolomide given to patients with *MGMT* promoter-methylated tumors also provided benefits, with a median OS of 21.7 months (95% CI, 17.4–30.4), compared to 15.3 months (95% CI, 13.0–20.9) in patients with unmethylated tumors (*p* = 0.007); no survival benefit with temozolomide addition was observed in patients with unmethylated tumors (Appendix A) [28]. A meta-analysis of phase 3 studies of treatments for newly diagnosed glioblastoma conducted between 2005 and 2022 further demonstrated improved OS in young patients [38]. The benefit of adding temozolomide to radiotherapy for patients with *MGMT* promoter methylation was confirmed in a recent meta-analysis of data from five randomized controlled trials (RCTs) [39]. However, there are conflicting data regarding the duration of adjuvant temozolomide. A large meta-analysis of pooled data from 2578 patients with newly diagnosed glioblastoma enrolled in 5 RCTs and 16 nonrandomized comparative studies showed that, compared with standard adjuvant temozolomide, extended adjuvant temozolomide was associated with a reduced risk of progression and death [40]. However, this association was significant only for patients enrolled in nonrandomized comparative studies, which have a high risk of bias [40]. In contrast, a recent meta-analysis involving patients with newly diagnosed glioblastoma showed that continuing adjuvant temozolomide beyond six cycles did not increase PFS or OS [41]. There was no consensus regarding the extension of maintenance therapy up to 12 months in patients who show a partial response or continuous improvement due to panelists reporting a lack of evidence.

For patients aged > 70 years with a good performance status (KPS ≥ 60), the panelists recommend 40 Gy hypofractionated radiotherapy plus concurrent temozolomide 75 mg/m^2^ for 21 days, followed by adjuvant temozolomide 150–200 mg/m^2^ in a 5/28-day schedule for 6–12 cycles. However, a reanalysis of the CE.6 and the pooled Nordic/NOA-08 trials suggests no benefit from temozolomide treatment for older (>60 years old) patients with glioblastoma with truly unmethylated *MGMT* promoter [42]. One of the panelists noted that these recent findings have changed the treatment of this population in Edmonton, a contrast with the current NCCN guidelines [43]. The panelists also reached a consensus regarding the use of hypofractionated radiotherapy with or without concurrent or adjuvant temozolomide, temozolomide alone (only in the presence of *MGMT* methylation), or palliative care alone in patients with a poor performance status (KPS < 60). Options for hypofractionated radiotherapy include 25 Gy in 5 fractions, 34 Gy in 10 fractions, or 40 Gy in 15 fractions with or without temozolomide [44,45,46]. These recommendations largely align with the most recent Canadian guidelines for glioblastoma management [12].

#### 4.2.4. New First-Line Treatments

The panelists reached a consensus regarding the combination of TTFields with adjuvant temozolomide after treatment with standard chemoradiotherapy for patients with a good performance status (KPS ≥ 60), regardless of *MGMT* promoter methylation status. Similarly, the 2024 clinical practice guidelines published by the US National Comprehensive Cancer Network (NCCN) include TTFields for patients with newly diagnosed glioblastoma who have a good performance score (KPS ≥ 60) [43]. According to the NCCN guidelines, TTFields may be used as an add-on treatment option to standard radiotherapy combined with concurrent and adjuvant temozolomide chemotherapy for tumors presenting with methylated, indeterminate, or unmethylated *MGMT* promoter in patients who are aged 70 years or older, as well as patients who are younger than 70 years [43].

In Canada, TTFields was approved in November 2022 for the management of glioblastoma [47]. This treatment approach uses a non-invasive wearable device that sends low-intensity electric fields via transducer arrays to the tumor site through the skin of the scalp [48]. Data from the phase 3 EF-14 trial suggest that TTFields improve OS and PFS in patients with newly diagnosed glioblastoma (including in patients ≥ 65 years old) when added to standard treatment with temozolomide chemotherapy following surgery and concurrent chemoradiotherapy using temozolomide (Appendix A) [49,50]. This open-label phase 3 study included 695 patients who were randomized to receive TTFields plus temozolomide or temozolomide alone, following standard radiochemotherapy [50]. Compared to temozolomide alone, TTFields plus temozolomide resulted in significant improvements in both PFS (6.7 months vs. 4.0 months; hazard ratio [HR], 0.63 [95% CI, 0.52–0.76]; *p* < 0.001) and OS (20.9 months vs. 16.0 months; HR, 0.63; [95% CI, 0.53–0.76]; *p* < 0.001) [50]. Consistently, a meta-analysis of seven trials comparing standard chemoradiotherapy to chemoradiotherapy combined with TTFields demonstrated that the addition of TTFields to standard chemoradiotherapy significantly improved OS in patients with newly diagnosed glioblastoma (median OS: 22.6 months [95% CI, 17.6–41.2] vs. 17.4 months [95% CI, 14.4–21.6]) [19]. Another meta-analysis of phase 3 trials showed that TTFields plus maintenance temozolomide also improved OS in young patients with newly diagnosed glioblastoma [38]. In these studies, the most common adverse event associated with TTFields was skin irritation, which was usually mild to moderate and treatable. Systemic adverse events were similar or lower with TTFields therapy compared to chemotherapy [49,50,51]. Finally, a preliminary report from a large perspective, non-interventional study showed results consistent with prior trials. After 56.2 months, the median PFS of patients who opted for TTFields (*n* = 429) was 10.2 months (95% CI, 9.4–11.4), and the median OS was 19.6 months (95% CI, 17.9–22.4) [52].

### 4.3. Monitoring Treatment Response and Disease Progression

Despite treatment with aggressive therapies, recurrence rates in patients with glioblastoma are high. Tumor recurrence is observed in approximately 90% of patients with glioblastoma 1–2 years after surgical tumor removal [5]. In approximately 50% of patients with tumor recurrence, tumor progression occurs in the wall of the resection cavity or within 2 cm of its margin [5,53]. The median time to recurrence after debulking surgery, followed by concomitant chemoradiotherapy, is approximately 9 months from initial diagnosis [54]. The panelists reached a consensus that tumor progression or recurrence should be monitored using gadolinium-enhanced MRI, and tumor recurrence should be determined according to the Response Assessment in Neuro-Oncology (RANO) Working Group criteria. This recommendation aligns with the most recent Canadian guidelines for glioblastoma management [11]. The panelists also reached a consensus on performing a brain MRI 4–6 weeks after radiotherapy, then every 2–4 months while on treatment, and subsequently at intervals determined by the physician.

Pseudoprogression (PsPD), which is transient radiological changes that mimic tumor progression, most likely due to treatment, is common in patients with glioblastoma undergoing treatment with chemoradiotherapy. According to a Canadian study of 111 patients, approximately one-third of patients with early progression after treatment with radiotherapy and temozolomide experience PsPD, rather than true tumor growth, which is associated with a more favorable prognosis [6]. Another study of Canadian patients treated with chemoradiotherapy showed an even higher incidence of PsPD, with 58% (25 of 43 evaluable patients) showing early MRI progression after concurrent chemoradiotherapy; these patients had better survival than those with true progression (median OS: 14.5 vs. 9.1 months; *p* = 0.025) [7]. Among the commonly used glioblastoma response criteria (e.g., RECIST, MacDonald, and RANO), RANO provides the lowest incidence of PsPD, according to data from 130 patients with glioblastoma treated with chemoradiotherapy at a single Canadian institution [8]. Considering the high incidence of PsPD after chemoradiotherapy, the panelists recommend that patients receiving chemoradiation should not be classified as having tumor progression based on gadolinium-enhanced MRI within the first 12 weeks after the end of radiotherapy, unless new enhancement is evident outside the radiotherapy field, or the presence of a viable tumor is confirmed by a pathologist at the time of reoperation. They also recommend the continuation of adjuvant temozolomide in patients with suspected PsPD. These recommendations to discriminate between PsPD and true progression largely align with previous recommendations [11]. Moreover, recognizing that PsPD may be difficult to distinguish from true progression based on clinical and radiological criteria, the panelists recommend the use of advanced imaging techniques, such as perfusion MRI, diffusion-weighted MRI, MR spectroscopy, and amino acid PET/CT, in correlation with conventional MRI findings. Advanced neuroimaging techniques, including diffusion-weighted MRI, dynamic susceptibility contrast perfusion MRI, 18F-fluoroethyltyrosine PET (^18^F-FET PET), and amide proton transfer-weighted MRI (APTw-MRI), indeed provided promising diagnostic accuracy in differentiating PsPD from true progression for 1372 patients with high-grade gliomas [18]. According to another recent meta-analysis of data from 21 studies, 18-fluorodihydroxyphenylalanine PET (^18^F-FDOPA PET) scans may also be useful in distinguishing between PsPD and true progression, although with moderate accuracy [55].

### 4.4. Management of Recurrent or Progressive Glioblastoma

#### 4.4.1. General

Second-line treatment for patients with progressive tumors or recurrent disease may vary, depending on the tumor’s response to the initial treatment. There is no acknowledged standard of care for patients with progressive glioblastoma. The panelists reached a consensus that a personalized treatment approach is needed for the management of recurrent or progressive glioblastoma, considering the tumor characteristics (tumor size, location, and molecular profile), the response to initial treatment, the impact of treatment on quality of life, and the patient’s age, functional ability (measured as KPS), symptoms, needs, and preferences. The experts also agreed that clinical trial participation should be encouraged for all patients with recurrent disease.

#### 4.4.2. Surgery

The panelists agreed that, in consultation with the multidisciplinary team and the patient, reoperation should be carefully considered for patients with recurrent glioblastoma, depending on factors such as tumor size, location, performance status, and time since previous treatment. They also reached a consensus that reoperation may benefit select patients with large, symptomatic, and resectable recurrent tumors, especially if the tumor has recurred after a long interval. A single-center study of 180 Canadian patients with recurrent glioblastoma confirmed that repeat surgery followed by additional salvage therapies could prolong OS in patients with recurrent glioblastoma (Appendix A) [56]. The median OS for patients who underwent repeat surgery was 9.6 months, compared to 5.3 months for patients who did not undergo reoperation (HR, 0.534 [95% CI, 0.393–0.939]; *p* < 0.0029) [56]. Additional retrospective and observational studies have shown a statistically significant OS benefit for repeat surgery in patients with recurrent glioblastoma [57,58,59]. However, caution is needed when interpreting the results of retrospective and observational studies, as only patients with a high-performance status undergo surgery and additional treatments, whereas those with poor performance status are offered only supportive therapy. Considering this evidence, the panelists reached a consensus that repeat surgery should be considered only for patients with a high-performance status score (KPS > 70) and those with a tumor in a favorable location until high-quality evidence on the benefit of repeat surgery and the optimal timing of reoperation becomes available.

#### 4.4.3. Radiotherapy and Chemotherapy

Re-irradiation has long been offered as a salvage therapy for recurrent glioblastoma and should be considered in select patients, such as those with a small tumor size, a favorable tumor location, a good performance status, and a long time since previous treatment. However, the panelists emphasized that re-irradiation is variably used in Canada and is center-specific. Observational studies suggest trends toward improvements in survival and disease control after re-irradiation (especially when combined with bevacizumab) [60,61,62,63]. Furthermore, the NRG Oncology/RTOG 1205 phase II prospective clinical trial that randomized 170 eligible patients with recurrent glioblastoma to bevacizumab alone versus re-irradiation (35 Gy in 10 fractions) and bevacizumab showed that re-irradiation provided a progression-free survival benefit, but no significant overall survival benefit [64].

The panelists agreed that lomustine and rechallenge with temozolomide may benefit some patients with recurrent glioblastoma. A recent real-world study involving 422 patients demonstrated that patients who received chemotherapy in combination with other treatments (surgery, radiotherapy, or both) for glioblastoma recurrence experienced longer OS than those who received palliative care (HR, 4.67; *p* < 0.001). However, patients receiving palliative care were also significantly older, had a worse KPS, and experienced a shorter time to recurrence [65]. According to a network meta-analysis of 42 studies (34 RCTs and eight nonrandomized studies) involving a total of 5236 patients, the evaluated combination treatments did not improve overall survival for patients with a first recurrence of glioblastoma compared with lomustine monotherapy [66]. Other chemotherapeutic agents and novel treatments failed to improve outcomes, or the evidence was uncertain [66]. Although chemotherapy may offer some benefits in the management of recurrent glioblastoma, its efficacy appears to be contingent on various factors, including the specific agents used, *MGMT* methylation status, and the context of other treatments [65].

The panelists agreed that bevacizumab, an antiangiogenic agent approved by Health Canada for the treatment of patients with glioblastoma after relapse or disease progression [67], may be added to chemotherapy for select patients. They also reached a consensus that bevacizumab may be combined with radiotherapy to mitigate the increased risk of radiation damage during secondary radiation in select patients with second recurrence. According to the aforementioned network meta-analysis involving a total of 5236 patients, radiotherapy with or without bevacizumab may play a role in the treatment of second recurrence [66]. The study also showed that the addition of bevacizumab to lomustine did not improve OS compared with lomustine alone, and the combination of bevacizumab and lomustine was associated with a high risk of serious side effects. Furthermore, data from meta-analyses suggested that, although bevacizumab may improve the quality of life and PFS of patients with newly diagnosed glioblastoma, its effect on OS remains inconclusive [68,69]. Other meta-analyses demonstrated that treatment with bevacizumab improved PFS and cognitive function in patients with recurrent glioblastoma, although the effects of bevacizumab on OS were inconsistent across studies [70,71,72]. The ability of bevacizumab to improve survival for patients with recurrent glioblastoma may be stronger when bevacizumab was combined with lomustine, radiotherapy, irinotecan, or temozolomide [70,72,73]. *IDH* mutation status, a large tumor burden, hyperintense lesions in T1, and diffusion-weighted restriction were associated with a response to bevacizumab [70]. Another meta-analysis of four studies involving 552 patients with recurrent high-grade glioma showed that reduced-dose bevacizumab provided similar benefits as standard-dose bevacizumab (5 mg/kg/week), with fewer side effects [74]. One retrospective Canadian study found that reduced-dose bevacizumab was associated with improved overall survival outcomes, suggesting that further exploration of optimal dosing is warranted [75]. Notably however, in the BELOB phase III clinical trial, no survival advantage was found in using bevacizumab and lomustine in combination at first recurrence of glioblastoma versus lomustine alone. As such, while bevacizumab is one of the few treatment options available for recurrent glioblastoma, its impact on overall survival has not been reproduced in all studies [76]. Considering this evidence, the panelists reached a consensus that bevacizumab may be added to chemotherapy in select patients.

#### 4.4.4. New and Emerging Treatments

The panelists did not reach a consensus regarding the usefulness of monotherapy with TTFields in patients with recurrent or progressive glioblastoma (consensus level: 67%) due to a lack of sufficient evidence. This contrasts with the NCCN recommendations (NCCN). There is some evidence that TTFields therapy alone may offer comparable efficacy to chemotherapy alone in patients with recurrent glioblastoma [77,78]. In a randomized phase 3 trial of 237 patients with recurrent glioblastoma (with only 12% at first recurrence), TTFields demonstrated comparable efficacy, but not superiority, to conventional chemotherapies [77]. The median OS was 6.6 months in the TTFields group and 6.0 months in the chemotherapy group (HR, 0.86 [95% CI, 0.66–1.12]; *p* = 0.27). Moreover, TTFields improved various domains of quality of life compared with chemotherapy [77]. TTFields-related adverse events were generally mild to moderate skin rashes, with severe adverse events occurring less frequently compared to chemotherapy. Furthermore, a post hoc analysis of the efficacy and safety of TTFields combined with second-line therapy after the first recurrence in patients enrolled in the phase 3 EF-14 trial showed a median OS of 11.8 months with second-line TTFields plus chemotherapy and 9.2 months with chemotherapy alone (HR, 0.70 [95% CI, 0.48–1.00]; *p* = 0.049) [78].

The experts agreed that targeted therapy and immunotherapy could be considered for the management of recurrence in patients with specific genetic alterations such as *BRAF* V600E mutation, *NTRK* fusions, and high microsatellite instability (MSI-H). Several meta-analyses of trials involving patients with recurrent glioblastoma showed trends toward improved OS and PFS in patients receiving immunotherapy; however, improvements in survival were statistically insignificant, and efficacy varied considerably among trials [79,80,81]. The phase 3 randomized CheckMate 143 trial investigated the efficacy of bevacizumab and the anti-PD-1 antibody nivolumab in 369 patients with recurrent glioblastoma. At a median follow-up of 9.5 months, OS was similar in the two groups (median OS: 10.0 months [95% CI, 9.0–11.8] for bevacizumab vs. 9.8 months (95% CI, 8.2–11.8) for nivolumab; *p* = 0.76) [82]. For patients with hypermutant tumors in the setting of replication repair deficiency syndromes (such as Lynch syndrome or constitutional mismatch repair deficiency), immunotherapy with checkpoint inhibition combined with re-irradiation may improve survival [83]. Moreover, therapies targeting tumor-specific mutations have shown promise in prolonging patient survival. A meta-analysis of 12 RCTs of targeted therapies (protein kinase inhibitors, proteasome and histone deacetylase inhibitors, antiangiogenic therapies, and poly-ADP-ribose polymerase inhibitors) showed that none of the targeted therapies significantly increased OS compared with the standard of care (radiotherapy plus temozolomide) [84]. Promising results of a phase 2 trial regarding regorafenib, a multikinase inhibitor, failed to be confirmed in a subsequent phase 3 trial [85,86]. Furthermore, the development of resistance to targeted therapy is common [4], and these treatments can benefit only patients harboring specific mutations. Various receptor tyrosine kinase (RTK) inhibitors are currently under clinical investigation in patients with glioblastoma, including agents targeting the EphA3 receptor, EGFR, and VEGF (Appendix A) [4]. However, the clinical success of most RTK inhibitors has been limited.

### 4.5. Supportive Care

Because glioblastoma affects brain function, many patients experience drastic deterioration in their mental health and quality of life [9,10]. Glioblastoma can cause significant disability, and many patients are unable to work or drive [10]. Epilepsy and cognitive changes affect multiple facets of caregiver quality of life and significantly contribute to increased caregiver burden [87]. The panelists agreed that treatment with antiepileptic drugs (e.g., levetiracetam, lacosamide, and perampanel) should be considered for patients who experience seizures.

The high disability burden in patients with glioblastoma may cause anxiety and depression in patients and caregivers, further compromising their quality of life and personal, social, and professional lives [10,87]. A recent meta-analysis of six studies involving patients with glioma showed that depression was significantly associated with poor survival outcomes, particularly among patients with high-grade glioma [88]. This correlation between depression and poor survival was significant both before and after surgery, highlighting the importance of early identification and intervention for depression in patients with glioblastoma, regardless of the phase in the disease trajectory. Meaning-centered psychotherapy has been shown to improve the quality of life of patients with glioblastoma and their caregivers [89]. However, the effects of psychological treatment on survival outcomes remain unclear. Considering this evidence, the experts reached a consensus that early identification of and intervention for depression among patients with glioblastoma may be clinically important in terms of glioblastoma outcomes, and patients with signs of depression should be referred to a specialist for evaluation and treatment. They also agreed that rehabilitation may be considered for select patients with glioblastoma. However, there was no consensus regarding the prophylaxis or management of VTE. Panelists indicated that these recommendations were out of scope.

### 4.6. Limitations

The limitations of this study include the potential for selection bias, as the experts were chosen based on their clinical expertise and research contributions, which may not fully reflect the diversity of clinicians treating patients with glioblastoma across Canada. Additionally, the consensus recommendations presented in this article rely on expert opinion, which can be influenced by individual biases and experiences. Another limitation is that the adoption of these recommendations across Canada may face various challenges, including resource constraints, inconsistent insurance coverage and personal financial limitation for medications and devices, and varying access to new treatments and advanced imaging modalities across provinces. For instance, during the Delphi process, some experts emphasized that advanced MRI techniques are subject to resource availability and that sensitivity and reproducibility are variable.

## 5. Conclusions

The management of glioblastoma remains a significant challenge, with a poor prognosis and high mortality rates. Although recent advances have been made in the diagnosis, classification, and treatment of glioblastoma, more work is needed to improve outcomes for Canadian patients. These consensus recommendations provide a comprehensive, evidence-based framework for the management of glioblastoma in Canada. They emphasize the importance of molecular diagnostics, multidisciplinary care, and personalized treatment approaches. Timely diagnosis using MRI and molecular testing is critical because glioblastoma is an aggressive and rapidly growing tumor. Incorporating molecular markers, such as *IDH* mutations and *MGMT* promoter methylation status, can aid in classification and prognosis. Although standard first-line treatment involves maximal safe surgical resection, followed by chemoradiation with temozolomide, TTFields have also emerged as a new treatment option in the first-line setting. Careful patient monitoring and distinction between true tumor progression and PsPD are essential. For recurrent disease, options include repeat surgery, chemotherapy (particularly lomustine-based regimens), re-irradiation, and antiangiogenic therapy with bevacizumab. However, evidence to guide optimal second-line management remains limited. Ongoing research is exploring novel targeted therapies, immunotherapies, and innovative trial designs to accelerate drug development for this aggressive disease. As our understanding of glioblastoma biology continues to evolve and new therapies emerge, regular updates to these recommendations will be crucial to ensure optimal patient care.

## Figures and Tables

**Figure 1 curroncol-32-00207-f001:**
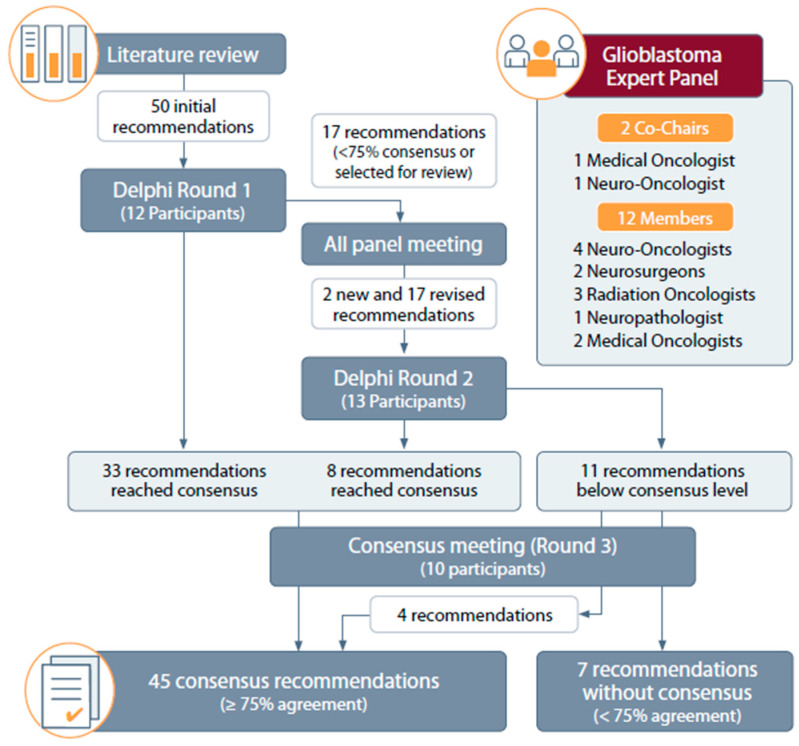
Flowchart of the Delphi consensus process.

**Figure 2 curroncol-32-00207-f002:**
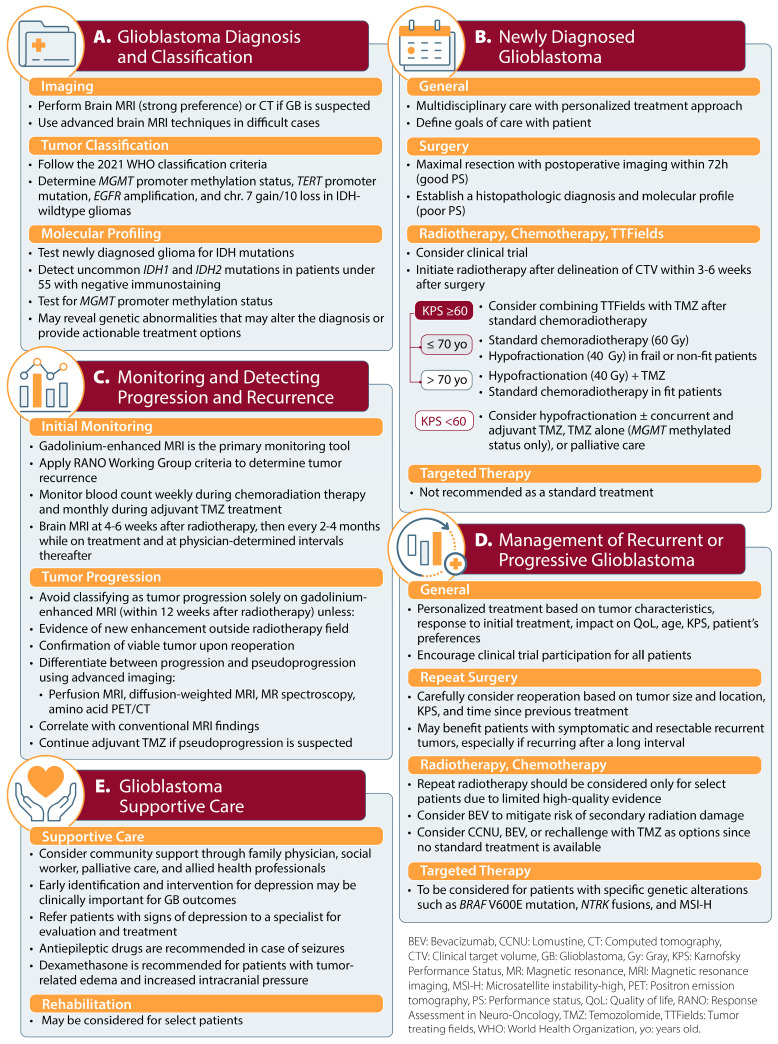
Recommendations for the diagnosis and management of glioblastoma.

**Table 1 curroncol-32-00207-t001:** Consensus statements related to glioblastoma diagnosis and classification ^a^.

Statement	Consensus Level	Consensus Strength	Evidence Quality ^b^	Round
**Imaging**				
All patients with a suspected diagnosis of GB should receive a brain MRI. The minimum assessment should include T2, FLAIR, and pre- and post-contrast T1 sequences.	100%	Strong	Moderate to high	1
CT should be used to evaluate patients with suspected GB only if MRI is unavailable or contraindicated.	92%	Strong	High	1
Advanced brain MRI techniques, such as diffusion-weighted MRI, perfusion-weighted MRI, PET, and MR spectroscopy, may be considered to assess the presence of GB in difficult cases (e.g., in individuals with intracranial hemorrhage) or to help distinguish GB from other tumor types and progression from radionecrosis.	92%	Moderate	Moderate	1
**Tumor Classification**				
IDH-wildtype grade 4 gliomas should be classified as GB in accordance with the 2021 WHO classification criteria.	77%	Strong	High	1
IDH-wildtype astrocytic gliomas in adults should be considered as GB if any of the following criteria are met: microvascular proliferation, necrosis, *TERT* promoter mutation, *EGFR* gene amplification, and +7/−10 chromosome copy number changes.	77%	Strong	High	1
*MGMT* promoter methylation and *TERT* promoter mutations should be determined in patients with IDH-wildtype diffuse gliomas.	85%	Moderate	Moderate	1
**Molecular Profiling**				
All patients with newly diagnosed gliomas should be tested for *IDH* mutations.	100%	Strong	High	1
Sequencing of *IDH1* and *IDH2* is recommended to detect uncommon *IDH1* and *IDH2* mutations in patients under 55 years of age with negative immunostaining, without postponing treatment.	100%	Strong	Moderate	2
In patients with confirmed GB (IDH-wildtype) or grade 4 IDH-mutated tumors, *MGMT* promoter methylation status should be tested to inform prognosis and predict response to temozolomide chemotherapy.	77%	Strong	Moderate	1
Analysis of *H3K27* mutations should be considered in patients with high-grade midline, IDH-wildtype gliomas.	100%	Strong	High	2
Key molecular markers of patients with GB include *TERT* promoter mutation, *EGFR* amplification, and chromosome 7 gain and 10 loss, without postponing treatment. Molecular profiling may reveal other genetic abnormalities that may alter the diagnosis or provide actionable treatment options.	75%	Moderate	Moderate	3

^a^ The consensus strength, level, and evidence quality are given for the round at which the statements reached the consensus level; ^b^ see Appendix A for details. CT: computed tomography; FLAIR: fluid-attenuated inversion recovery; GB: glioblastoma; MR: magnetic resonance; MRI: magnetic resonance imaging; PET: positron emission tomography.

**Table 2 curroncol-32-00207-t002:** Consensus statements related to the management of newly diagnosed glioblastoma ^a^.

Statement	Consensus Level	Consensus Strength	Evidence Quality ^b^	Round
**General**				
A multidisciplinary approach is required for GB management. Multidisciplinary care teams would optimally include neurosurgeons, medical oncologists or neuro-oncologists, radiation oncologists, neuropathologists, radiologists, and neurologists.	92%	Strong	Moderate	1
Follow a personalized treatment approach for the management of newly diagnosed GB patients, considering the tumor characteristics (size, location, and molecular profile) and the patient’s age, functional ability (measured as KPS), symptoms, clinical needs, and preferences.	100%	Strong	High	1
Given that the intent of management is palliative from the outset, part of the discussion on further management should involve a conversation with the patient and their family about goals of care and whether they wish to proceed with active therapies. For some patients with a very poor performance status, palliative care alone is a reasonable option.	92%	Moderate	High	1
**Surgery**				
Maximal safe resection is recommended for all patients with newly diagnosed GB when the tumors are operable and for all patients who have good performance status. In cases not amenable to resection (i.e., poor performance status and/or those with unfavorable tumor location), stereotactic biopsy is recommended to establish a histopathologic diagnosis and molecular profile (including cytogenetic alterations). Whenever feasible, tissue should be saved for tumor banking.	83%	Moderate-to-strong	Moderate	1
Advanced techniques, such as fluorescence-guided surgery, may be used to optimize tumor removal and preserve normal brain tissue.	75%	Moderate	Low	3
Postoperative imaging within 72 h is recommended for patients undergoing surgery for newly diagnosed GB.	100%	Strong	Moderate	1
**Radiotherapy, Chemotherapy, and TTFields**				
Where possible, patients should be considered for a clinical trial.	100%	Strong	High	1
Radiation therapy should be initiated as soon as it is safely permissible, ideally within 3–6 weeks after surgery.	92%	Moderate	Moderate	1
Use gadolinium-enhanced T1-weighted MRI and FLAIR to determine the CTV.	92%	Strong-to-moderate	High-to-moderate	2
Consider combining TTFields with adjuvant temozolomide after treatment with standard chemoradiotherapy for patients who have good performance status (KPS ≥ 60), regardless of *MGMT* promoter methylation status.	85%	Strong	High-to-moderate	2
For patients aged >70 years with good performance status (KPS ≥ 60), consider 40 Gy hypofractionated radiotherapy plus concurrent temozolomide 75 mg/m^2^ for 21 days, followed by adjuvant temozolomide 150–200 mg/m^2^ in a 5/28-day schedule for 6–12 cycles. Standard radiotherapy may also be considered in place of hypofractionated radiotherapy in fit patients.	85%	Moderate	Moderate	2
For patients aged ≤ 70 years with a good performance status (KPS ≥ 60), consider 60 Gy radiotherapy plus concurrent temozolomide 75 mg/m^2^ for 42 days, followed by adjuvant temozolomide 150–200 mg/m^2^ in a 5/28-day schedule for six cycles. Hypofractionated radiotherapy may also be considered in place of standard radiotherapy if patients are ineligible for chemotherapy, frail, or present significant comorbidities.	75%	Strong	High	3
In patients with poor performance status (KPS < 60), consider hypofractionated radiotherapy with or without concurrent or adjuvant temozolomide, temozolomide alone (only in the presence of *MGMT* methylation), or palliative care alone.	100%	Moderate	Moderate	2
**Targeted Therapy and Immunotherapy**				
Targeted therapy and immunotherapy have not shown clear survival benefits in patients with newly diagnosed GB and are not recommended as the standard of care for this population.	92%	Moderate	Moderate	1

^a^ The consensus strength, level, and evidence quality are given for the round at which the statements reached the consensus level; ^b^ see Appendix A for details. CTV: clinical target volume; FLAIR: fluid-attenuated inversion recovery; GB: glioblastoma; KPS: Karnofsky performance status; MRI: magnetic resonance imaging; TTFields: tumor treating fields.

**Table 3 curroncol-32-00207-t003:** Consensus statements related to patient monitoring and the detection of glioblastoma progression and recurrence ^a^.

Statement	Consensus Level	Consensus Strength	Evidence Quality ^b^	Round
Tumor progression or recurrence should be monitored using gadolinium-enhanced MRI, and tumor recurrence should be determined according to the RANO Working Group criteria.	75%	Moderate	Moderate	1
Blood counts should be monitored weekly during chemoradiation therapy and monthly during adjuvant temozolomide treatment.	83%	Strong	Moderate	1
Brain MRI is recommended 4–6 weeks after radiotherapy and then every 2–4 months while on treatment and at physician-determined intervals thereafter.	75%	Strong	Moderate	3
Patients receiving chemoradiation should not be classified as having tumor progression based on gadolinium-enhanced MRI within the first 12 weeks after the end of radiotherapy unless new enhancement is evident outside the radiotherapy field or the presence of a viable tumor is confirmed by a pathologist at the time of reoperation.	100%	Moderate-to-strong	Moderate	1
Adjuvant temozolomide should be continued in patients with suspected pseudoprogression.	92%	Strong	Moderate	1
In patients with suspected progressive GB, pseudoprogression may be difficult to distinguish from true progression based on clinical and radiological criteria. Therefore, advanced imaging techniques, such as perfusion MRI, diffusion-weighted MRI, MR spectroscopy, and amino acid PET/CT, can be used in correlation with conventional MRI findings to distinguish pseudoprogression from true progression.	83%	Moderate	Low-to-moderate	1

^a^ The consensus strength, level, and evidence quality are given for the round at which the statements reached the consensus level; ^b^ see Appendix A for details. CT: computed tomography; GB: glioblastoma; MR: magnetic resonance; MRI: magnetic resonance imaging; PET: positron emission tomography; RANO: Response Assessment in Neuro-Oncology.

**Table 4 curroncol-32-00207-t004:** Consensus statements related to the management of recurrent or progressive glioblastoma ^a^.

Statement	Consensus Level	Consensus Strength	Evidence Quality ^b^	Round
**General**				
Follow a personalized treatment approach for the management of recurrent or progressive GB, considering the tumor characteristics (tumor size, location, and molecular profile), the response to initial treatment, the impact of treatment on the patient’s quality of life, and the patient’s age, functional ability (measured as KPS), symptoms, needs, and preferences.	100%	Strong	Moderate	1
**Clinical Trials**				
Clinical trial participation should be encouraged for all patients with recurrent GB.	100%	Strong	Moderate	1
**Surgery**				
In consultation with the multidisciplinary team and the patient, reoperation should be considered for patients with recurrent GB, depending on factors such as tumor size, location, performance status, and time since previous treatment.	100%	Moderate-to-strong	Moderate	1
Reoperation may benefit select patients with large, symptomatic, and resectable recurrent tumors, especially if the tumor has recurred after a long interval.	100%	Strong	Moderate	1
Reoperation may provide some survival benefits for select patients, but it also carries risks of complications. Repeat surgery should be considered only for patients with a high-performance status score (KPS > 70) and those with a tumor in a favorable location until high-quality evidence on the benefit of repeat surgery and the optimal timing of reoperation becomes available.	83%	Moderate	Moderate	1
**Radiotherapy, Chemotherapy, and TTFields**				
There is no high-quality evidence to support the use of repeat radiotherapy in patients with recurrent GB, and re-irradiation should be considered only for select patients with recurrent GB, such as those with a small tumor size, a favorable location, and a good performance status, as well as those for whom a long time has passed since previous treatment.	83%	Moderate	Moderate	1
Radiotherapy with bevacizumab to mitigate the increased risk of radiation damage during secondary radiation may be considered for the treatment of select patients with second recurrence.	75%	Moderate	Low-to-moderate	1
There is no standard treatment for recurrent GB. Options include lomustine and rechallenge with temozolomide. Bevacizumab may be added to chemotherapy with select patients.	75%	Moderate	Moderate	1
**Targeted Therapy and Immunotherapy**				
The use of targeted therapy and immunotherapy could be considered for the management of recurrence in patients with specific genetic alterations such as *BRAF* V600E mutation, *NTRK* fusions, and microsatellite instability—high (MSI-H).	92%	Moderate	Moderate	2

^a^ The consensus strength, level, and evidence quality are given for the round at which the statements reached the consensus level; ^b^ see Appendix A for details. GB: glioblastoma; KPS: Karnofsky performance status; TTFields: tumor-treating fields.

**Table 5 curroncol-32-00207-t005:** Consensus statements related to supportive care ^a^.

Statement	Consensus Level	Consensus Strength	Evidence Quality ^b^	Round
Multidisciplinary care for GB may include community support through the family physician, a social worker, palliative care, and allied health professionals.	100%	Strong	Moderate	2
Early identification and intervention for depression in patients with GB may be clinically important in terms of GB outcomes, and patients with signs of depression should be referred to a specialist for evaluation and treatment.	83%	Moderate	Moderate	1
Treatment with antiepileptic drugs (e.g., levetiracetam, lacosamide, and perampanel) is recommended for patients with seizures.	100%	Strong	High	1
Treatment with dexamethasone is recommended for patients with tumor-related edema and increased intracranial pressure.	83%	Strong	Moderate	1
Rehabilitation may be considered for select patients with GB.	83%	Moderate	Moderate	1

^a^ The consensus strength, level, and evidence quality are given for the round at which the statements reached the consensus level; ^b^ see Appendix A for details. GB: glioblastoma.

**Table 6 curroncol-32-00207-t006:** Recommendations that did not reach consensus.

Statement	Consensus Status
**Management of Newly Diagnosed GB**	
The use of laser-interstitial thermal therapy, awake craniotomy, and functional MRI may be an option for patients with deep-seated inoperable tumors.	Did not reach consensus (consensus level: 46%)
For patients aged ≤ 70 years with good performance status (KPS ≥ 60), maintenance therapy could be continued for up to 12 months among patients who show continued benefit, improvement, or a partial response.	Did not reach consensus (consensus level: 46%)
**Management of Recurrent or Progressive GB**	
Consider monotherapy with TTFields for patients with recurrent or progressive GB.	Did not reach consensus (consensus level: 67%)
**Supportive Care**	
VTE prophylaxis with low-molecular-weight heparin is recommended within 24 h of surgical tumor resection for a minimum of seven days.	Did not reach consensus (consensus level: 46%)
Patients who develop VTE in the course of their illness should be treated with either low-molecular-weight heparin or direct oral anticoagulants for a duration determined on an individualized basis.	Did not reach consensus (consensus level: 57%)
Prophylaxis for Pneumocystis pneumonia is recommended for patients receiving chemoradiotherapy or adjuvant chemotherapy with temozolomide.	Did not reach consensus (consensus level: 23%)
TTFields prescriptions for patients with GB should be provided under the guidance of a supervising oncologist.	Did not reach consensus (consensus level: 62%)

GB: glioblastoma; KPS: Karnofsky performance status; MRI: magnetic resonance imaging; TTFields: tumor-treating fields; VTE: venous thromboembolism.

## Data Availability

No new data were created or analyzed in this study. Data sharing is not applicable to this article.

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
