# Peer review of "Canadian Expert Consensus Recommendations for the Diagnosis and Management of Glioblastoma: Results of a Delphi Study"

_curroncol, 2025, doi:10.3390/curroncol32040207_

Round 1
Reviewer 1 Report
Comments and Suggestions for Authors
Mason et al. report a Delphi study that aimed to develop Canadian consensus recommendations for the diagnosis, classification, and management of newly diagnosed and recurrent glioblastoma. The study was rigorously conducted and well presented. However, there are a couple of areas of improvement the authors should consider before publication:
- How do these results/recommendations differ from those presented by colleagues in other countries? This study and its impact relies on an implied difference/uniqueness of the recommendations in Canada compared to other countries as such this should be discussed further highlighting specific areas where these recommendations break from those given in USA or European countries
- Further discussion on the statements that did not reach consensus would be helpful to increase the novelty and interest in this article. Particularly why certain statements failed to reach consensus and expanding further, what new information/data would be required before consensus could be met on a particular treatment or diagnostic measure.
Author Response
Comments 1: How do these results/recommendations differ from those presented by colleagues in other countries? This study and its impact relies on an implied difference/uniqueness of the recommendations in Canada compared to other countries as such this should be discussed further highlighting specific areas where these recommendations break from those given in USA or European countries
Response 1: Thank you for your comment. These recommendations update the previous Canadian recommendations published in 2010 and 2017. They are evidence-based and designed to be applicable within the Canadian healthcare system. Generating unique recommendations was not in the design of the study. While the core of these recommendations is fundamentally similar to that of the NCCN guidelines, in the US, and the EANO guidelines, in Europe, there are notable differences. For examples, the recommendations include bevacizumab (not approved in Europe for glioblastoma); omit TTFields for recurrent glioblastoma (recommended by the NCCN and EANO guidelines); suggest using temozolomide alone in patients with poor performance status, only in the presence of MGMT methylation (no limit about methylation status within the NCCN and EANO guidelines); and consider temozolomide for all patients with good performance status (unlike NCCN, which allows for standard or hypofractionated radiotherapy alone). The lack of established standard of care for patients with recurrent or progressive glioblastoma further adds to the difference. The manuscript was updated to highlight some of the differences (lines 366, 551-552) - Please note that the line numbers may vary based on the track change options selected in Words.
Comments 2: Further discussion on the statements that did not reach consensus would be helpful to increase the novelty and interest in this article. Particularly why certain statements failed to reach consensus and expanding further, what new information/data would be required before consensus could be met on a particular treatment or diagnostic measure.
Response 2: Clarifications were added for statements that did not reach consensus. Some panelists considered there was insufficient evidence (e.g., maintenance up to 12 months, TTFields monotherapy for patients with recurrent of progressive glioblastoma) while others felt the recommendations were out of scope (prophylaxis and management of VTE). The corresponding statements were expanded accordingly (lines 356-358, 551, 615).
Reviewer 2 Report
Comments and Suggestions for Authors
This is a comprehensive overview of clinical consensus of GBM diagnosis and treatment. I have some minor suggestions:
1. A brief introduction of TTFields is recommended to be added since it is an emerging physical cancer therapy.
2. A group from Germany has reported a promising efficacy of TTFields for newly diagnosed GBM. This should be added in the discussion part. (doi.org/10.1200/JCO.2024.42.16_suppl.2036)
Author Response
Comments 1: Thank you for your comment. A brief introduction of TTFields is recommended to be added since it is an emerging physical cancer therapy.
Response 1: A brief introduction of the TTFields was added to indicate it is a non-invasive treatment using of a wearable device that sends low-intensity electric fields via transducer arrays to the tumour site through the skin of the scalp (lines 387-389) - Please note that the line numbers may vary based on the track change options selected in Words.
Comments 2: A group from Germany has reported a promising efficacy of TTFields for newly diagnosed GBM. This should be added in the discussion part. doi.org/10.1200/JCO.2024.42.16_suppl.2036)
Response 2: The results from this large perspective study were added as suggested (lines 407-410).
Reviewer 3 Report
Comments and Suggestions for Authors
This study presented Canadian Expert Consensus Recommendations for the Diagnosis and Management of Glioblastoma. Presentation was good, and overall merit of the study is high. Limitation of the study was adequately provided.
Author Response
No comments